# Protective Effect of Resveratrol against Ischemia-Reperfusion Injury via Enhanced High Energy Compounds and eNOS-SIRT1 Expression in Type 2 Diabetic Female Rat Heart

**DOI:** 10.3390/nu11010105

**Published:** 2019-01-06

**Authors:** Natacha Fourny, Carole Lan, Eric Sérée, Monique Bernard, Martine Desrois

**Affiliations:** 1Aix-Marseille University, CNRS, Centre de Résonance Magnétique Biologique et Médicale (CRMBM), Faculté de Médecine, 27 Boulevard Jean Moulin, 13385 Marseille, CEDEX 05, France; carole.lan@univ-amu.fr (C.L.); monique.bernard@univ-amu.fr (M.B.); martine.desrois@univ-amu.fr (M.D.); 2Aix-Marseille University, INSERM, INRA, Centre de Recherche en Cardiovasculaire et Nutrition (C2VN), Faculté de Médecine, 27 Boulevard Jean Moulin, 13385 Marseille, CEDEX 05, France; seree.eric@gmail.com

**Keywords:** resveratrol, type 2 diabetes, ischemia-reperfusion, cardiac function, energy metabolism, mitochondria, endothelial function

## Abstract

Type 2 diabetic women have a high risk of mortality via myocardial infarction even with anti-diabetic treatments. Resveratrol (RSV) is a natural polyphenol, well-known for its antioxidant property, which has also shown interesting positive effects on mitochondrial function. Therefore, we aim to investigate the potential protective effect of 1 mg/kg/day of RSV on high energy compounds, during myocardial ischemia-reperfusion in type 2 diabetic female Goto-Kakizaki (GK) rats. For this purpose, we used ^31^P magnetic resonance spectroscopy in isolated perfused heart experiments, with a simultaneous measurement of myocardial function and coronary flow. RSV enhanced adenosine triphosphate (ATP) and phosphocreatine (PCr) contents in type 2 diabetic hearts during reperfusion, in combination with better functional recovery. Complementary biochemical analyses showed that RSV increased creatine, total adenine nucleotide heart contents and citrate synthase activity, which could be involved in better mitochondrial functioning. Moreover, improved coronary flow during reperfusion by RSV was associated with increased eNOS, SIRT1, and P-Akt protein expression in GK rat hearts. In conclusion, RSV induced cardioprotection against ischemia-reperfusion injury in type 2 diabetic female rats via increased high energy compound contents and expression of protein involved in NO pathway. Thus, RSV presents high potential to protect the heart of type 2 diabetic women from myocardial infarction.

## 1. Introduction

Cardiovascular (CV) complications are the first causes of morbidity and mortality in type 2 diabetic patients, particularly in women [1]. Cardio-protection is widely recognized in women, and surprisingly, is suppressed with type 2 diabetes, with more serious CV consequences in women than in men [2]. It is known that the risk of myocardial infarction is five times higher in type 2 diabetic women compared with non-diabetic women, while this risk is only multiplied by two in men [3,4]. In addition, mortality due to myocardial infarction is higher in women than in men in type 2 diabetes [5]. Few studies explore female gender and the reasons for higher deterioration of the cardiovascular system are not yet fully understood. Endothelial damage is one likely hypothesis for CV complications in type 2 diabetes. Interestingly, Desrois et al. reported a higher endothelial damage in female GK rat hearts than in males in the absence of ischemic insult, which could explain the higher risk of CV in type 2 diabetic women [6].

Although this is one of the current objectives, most of antidiabetic treatments fail to decrease the CV risk [7]. For this purpose, dietary supplements could be interesting in combination with existing antidiabetic medication to improve CV outcomes in diabetic patients. Resveratrol (RSV), or trans-3,5,4′-trihydroxy stilbene, is a natural polyphenol found in more than 70 plant species like grapes, peanuts, and blackberries [8]. This molecule has shown pleiotropic and beneficial effects on both type 2 diabetes and cardiovascular complications [9]. Various studies, using models of type 2 diabetes or metabolic syndrome, showed that RSV could decrease chronic inflammation [10], improve insulin sensitivity [11], lipid profile [12], and decrease oxidative stress [13]. Other studies also demonstrated the effects of RSV on endothelial function mainly through mechanisms involving nitric oxide (NO) and sirtuin pathways [14,15,16,17]. Interestingly, RSV has shown beneficial effects on mitochondrial function by increasing mitochondrial DNA and biogenesis [13]. Consequently, RSV could be an interesting candidate to improve cardiac energy metabolism and CV outcomes of type 2 diabetic women [18,19,20].

In the literature, the dose of RSV is very different depending on the study. Doses range from 0.1 mg/kg/day [18] to 500 mg/kg/day [21] and even 4g/kg/day [22]. Here we choose a “low-dose” of RSV at 1 mg/kg/day based on Rocha et al. and Lin et al.’s studies. Indeed, Rocha et al. [23] chose a dose of 1 mg/kg based on the actual wine consumption in occidental countries, and the kinetics and bioavailability of resveratrol in the body. Lin et al.’s [18] study showed that a lower dose (0.1 mg/kg) was not sufficient to induce beneficial effects on the heart, while a dose of 1 mg/kg/day improved cardiac function. On the other hand, higher doses (25 mg/kg/day) have shown negative effects on infarct size [24]. Thus, we suppose that the “low-dose” of 1 mg/kg/day of RSV will be sufficient to observe beneficial effects on myocardial energy metabolism.

Here, we aim to determine the effects of a low-dose oral administration of RSV on myocardial sensitivity to ischemia-reperfusion injury in female GK rats, a polygenic model of type 2 diabetes, with the hypothesis that RSV could improve high energy compound contents. We believe this is an original study exploring the effects of RSV on high-energy compounds during an ischemia-reperfusion injury, using ^31^P magnetic resonance spectroscopy (MRS) and biochemical analysis, combined with measurement of myocardial function and coronary flow. Secondarily, we assessed the effects of RSV on coronary flow and expression of proteins involved in NO pathway, as indicators of endothelial function.

## 2. Materials and Methods

### 2.1. Materials and Antibodies

Assay kits were used to determine plasma glucose (Randox Laboratories, Crumlin, Antrim, UK) and free fatty acids (FFAs) (NEFA kit; Roche Diagnostics, Roche Applied Science, Mannheim, Germany). A radioimmunoprecipitation assay buffer (RIPA) lysis buffer was used to extract proteins (sc-24948, Santa Cruz Biotechnology, Santa Cruz, CA, USA). Total protein concentration was determined using the Pierce BCA protein assay kit (ref 23227, Thermo scientific, Rockford, USA). Anti-eNOS (ref 610296, BD Transduction Laboratories, USA), anti-SIRT1 (ref 9475, Cell Signaling Technology), anti-Akt (Cell Signaling Technology, Danvers, MA, USA), anti-PAkt (Ser 473) (Cell Signaling Technology, Danvers, MA, USA), anti-SIRT3 (#2627, Cell Signaling Technology, Danvers, MA, USA), and anti-Actin (sc47 778, Santa Cruz Biotechnology, Santa Cruz, CA, USA) primary antibodies were used for western blots. HRP-conjugated antibodies were used as secondary antibodies (Goat anti-mouse sc2031 or Goat anti-rabbit sc2030, Santa Cruz Biotechnology, Santa Cruz, CA, USA). The immunoblots were developed using an ECL Western Blotting Detection Reagent (GE Healthcare, AmershamTM, Buckinghamshire, U.K.). The protein signals were assessed using the MicroChemi 4.2 System (DNR Bio-Imaging System Ltd., Jerusalem, Israel). Citrate synthase activity was evaluated using the citrate synthase assay kit (CS0720, Sigma-Aldrich, St. Louis, MO, USA). First, protein extraction was performed using the CelLytic MT extraction buffer (C3228, Sigma, St. Louis, MO, USA). Malondialdehyde (MDA) was assessed with the lipid peroxidation assay kit (MAK085, Sigma-Aldrich, St. Louis, MO, USA).

### 2.2. Animals

Age-matched (7–8 months) female control Wistar rats (Charles River, France) and type 2 diabetic female Goto-Kakizaki (GK) rats (GK/Par subline; Laboratoire de Biologie et Pathologie du Pancréas Endocrine UMR8251-CNRS—Université Paris Diderot, Paris, France [25]) were used in the experiments. All procedures involving animals were approved by the Animal Experiment Ethics Committee of Aix-Marseille University (n°2017070416019134) and were in conformity with the European Convention for protection of animals used for experimental purpose. The animals were housed in a temperature controlled ventilated cabinet (22–24 °C) and were exposed to light–dark cycles of 12:12 h. Animals had access to food (diet 113, SAFE, Augy, France) and water ad libitum. Four groups were designed for this study: the control group (CTRL; *n* = 11), the type 2 diabetic group (GK; *n* = 14), the type 2 diabetic group under placebo treatment (GK-P; *n* = 9), and the type 2 diabetic group with RSV treatment (GK-RSV; *n* = 8).

### 2.3. Treatment

RSV was provided for 8 weeks in drinking water at the dose of 1 mg/kg/day as suggested before by Rocha et al. [23,26]. As RSV solubility is higher in ethanol, we first dissolved RSV in ethanol and then in water. The placebo treatment corresponded to 1‰ ethanol in drinking water. Daily water ingestion was evaluated a few weeks before the beginning of the study, to calculate the concentration of RSV solution. During the 8 weeks of RSV treatment, water consumption was also measured to adjust RSV concentration if necessary [23].

### 2.4. Myocardial Tolerance to Ischemia-Reperfusion Injury

After 8 weeks of RSV treatment, isolated perfused heart experiments were performed to evaluate ex vivo the tolerance to ischemia-reperfusion injury, by measuring energy metabolism, cardiac function and coronary flow during the whole protocol. As previously described, rats were anesthetized by intraperitoneal injection of 90 mg/kg pentobarbital sodium [27]. The hearts were quickly removed from the chest cavity by thoracotomy and arrested in ice-cold Krebs-Henseleit buffer (containing (mM): NaCl (118), KCl (4.7), MgSO_4_ (1.2), CaCl_2_ (1.75), NaHCO_3_ (25), KH_2_PO_4_ (1.2), EDTA (0.5) and D-glucose (11)). Hearts were weighed and then cannulated via the ascending aorta for retrograde Langendorff-perfusion of coronary arterial network at a constant pressure of 100 mm Hg. A drain was placed at the apex of the heart to evacuate coronary effluents. In the same time, blood samples were immediately taken for glucose and free fatty acids (FFAs) determination in plasma.

#### 2.4.1. Experimental Protocol

After 4 min of stabilization with a Krebs–Henseleit buffer, hearts were perfused for 24 min with a physiological recirculating Krebs–Henseleit buffer (Pa 0.4) containing 0.4 mM palmitate, 3% albumin, 11 mM glucose, 3U/L insulin, 0.8 mM lactate, and 0.2 mM pyruvate. Four minutes before low-flow ischemia, hearts were perfused with a physiological non-recirculating Krebs–Henseleit buffer (Pa 1.2) containing 1.2 mM palmitate, 3% albumin, 11 mM glucose, 3U/L insulin, 0.8 mM lactate, and 0.2 mM pyruvate. Then, the hearts underwent a low-flow ischemia (0.5 mL/min/g wet wt) of 32 min with the same buffer. Finally, flow was restored entirely for 32 min with the physiological Krebs–Henseleit buffer containing 0.4 mM palmitate. The palmitate concentration was increased at the end of the control period and during ischemia to induce a maximum damage [28]. The perfusates were continually gassed with a mixture of 95% O_2_ and 5% CO_2_ to maintain pH at 7.40. The buffer temperature was maintained at 37 °C during the entire protocol.

#### 2.4.2. Myocardial Function

A water-filled latex balloon was inserted in the left ventricle via the mitral valve and inflated to produce an end diastolic pressure (EDP) of ≈10 mm Hg at the beginning of perfusion. Left ventricular developed pressure (DP) and heart rate (HR) were recorded using a pressure sensor connected to the balloon, as previously described [6]. The product of heart rate and developed pressure was used as an index of cardiac function. During reperfusion, we calculated the percentage recovery between the pre-ischemic and post-ischemic cardiac function. Coronary flow (CF) was measured via collection of coronary effluent before and after ischemia (at 20 min and 80 min), expressed in mL/min/g wet weight.

#### 2.4.3. Myocardial Energy Metabolism

##### ^31^P Magnetic Resonance Spectroscopy (MRS)

Perfused rat hearts were placed in a 20-mm magnetic resonance sample tube and inserted in a ^31^P probe that was seated in the bore of a superconducting wide-bore (89-mm) 4.7 Tesla magnet (Oxford instruments, Oxford, U.K.) interfaced with a Bruker-Nicolet Avance WP-200 spectrometer (Bruker, Karlsruhe, Germany). ^31^P spectra were obtained by accumulating 328 free induction decay signals acquired for 4 min (flip angle 45°, repetition time 0.7 s, spectral width 4500 Hz, 2048 data points) [29]. Prior to Fourier transformation, the free induction decay was multiplied by an exponential function which generated a 20 Hz line broadening. Quantification of the signal integrals was carried out using an external reference containing an aqueous solution of 0.6 mM phenylphosphonic acid. A series of eight ^31^P spectra were recorded during each period of the experimental protocol to quantify phosphorus metabolites (ATP, PCr, and Pi) and intracellular pH.

##### Biochemical Analyses in Freeze-Clamped Heart

As a complement to ^31^P MRS, high performance liquid chromatography (HPLC) analysis, as well as citrate synthase (CS) activity, were performed as indicators of mitochondrial function. First, PCr, creatine, adenine nucleotides, and derivatives were assessed using ion-exchange high performance liquid chromatography (HPLC). A perchloric extraction, adapted from Lazzarino et al., was performed by homogenizing cardiac tissue (50 to 100 mg) with a Polytron homogenizer (Kinematica, Luzern, Switzerland) in ice-cold 0.6 M perchloric acid [30]. Then, homogenates were centrifuged at 5000× g for 10 min at 4 °C and supernatants were preserved for the comparative metabolite determination. Protein concentration calculation was carried out according to Lowry et al. to express the results in µmol/g protein [31]. Separation of adenine nucleotide derivatives, phosphocreatine, and creatine was performed using the ion-pairing reverse phase technique. Qualitative and quantitative analyses were carried out using adenine nucleotide standards and thymine monophosphate (Sigma, Poole, Dorset, UK) as an internal standard. Under these chromatographic conditions, a highly resolved separation of ATP, ADP, AMP, PCr, and creatine was obtained in 40 min. Total adenine nucleotide pool (TAN) was calculated from the sum ATP + ADP + AMP. Energy charge (EC) is equal to ((ATP + 0.5ADP) / (ATP + ADP + AMP)) × 10.

Secondly, CS activity was evaluated using the citrate synthase assay kit. Protein extraction was performed, extracts were centrifuged at 14,000× g for 10 min at 4 °C and total protein concentration in the supernatant was determined using the Pierce BCA protein assay kit. Activity of citrate synthase was assessed at 412 nm in a 96-well plate with a kinetic program. Results are expressed in nmol/g of protein/minute.

#### 2.4.4. Expression of Proteins Involved in NO Pathway

Complementary to coronary flow measurement, we assessed the expression of eNOS, SIRT1, Akt, and P-Akt proteins in freeze-clamped hearts. A piece of left ventricle tissue (≈60 mg) was homogenized in a RIPA lysis buffer and centrifuged at 14,000 rpm for 15 min at 4 °C. Total protein concentration in the supernatant was determined using the Pierce BCA protein assay kit. Equal amounts of proteins (90 µg for eNOS and SIRT1, 50 µg for Akt and P-Akt) were separated by 8% or 10% polyacrylamide gel electrophoresis and transferred onto nitrocellulose membranes. After blocking with 5% skim milk, membranes were incubated overnight at 4 °C with eNOS (1/1000), SIRT1 (1/1000), Akt (1/1000), P-Akt (Ser 473) (1/500), or Actin (1/2000) primary antibodies. Second, membranes were incubated with HRP-conjugated antibodies (1/2000). The immunoblots were developed and the protein signal was quantified using the Quantiscan software (Biosoft, Cambridge, U.K.). The intensity of each protein signal was normalized to the corresponding β-actin stain signal. Data are expressed as ratios between the protein and the corresponding β-actin signal density, except for P-Akt, which was expressed according to Akt.

#### 2.4.5. Oxidative Stress

SIRT3 protein expression, a mitochondrial sirtuin involved in oxidative stress, was assessed in freeze-clamped hearts following the same protocol as described above, with 50 µg of protein separated by 10% polyacrylamide gel. Primary antibody against SIRT3 was used at 1/1000.

MDA was assessed to evaluate lipid peroxidation in freeze-clamped hearts. Lipid peroxidation was determined by the reaction of MDA with thiobarbituric acid (TBA) to form a colorimetric (532 nm)/fluorometric (λex = 532/λem = 553 nm) product, proportional to the MDA present.

### 2.5. Statistical Analyses

Data are graphically provided as means ± SEM of absolute values. GraphPad Prism software 5.0 (La Jolla, CA, USA) was used for all statistical processing. Significant differences between groups were determined using two-way analysis of variance (ANOVA) with repeated measures over time for the time-dependent variables followed by Bonferroni post-hoc test. An unpaired Student’s *t*-test was used for the other parameters. A *p*-value of less than or equal to 0.05 was considered to indicate significant difference.

## 3. Results

### 3.1. Effect of Resveratrol on Physiological Parameters

Physiological parameters are shown in Table 1. Plasma glucose was significantly increased in GK, GK-P, and GK-RSV in comparison to CTRL (*p* < 0.0001). RSV treatment did not reduce plasma glucose in GK rats. Plasma-free fatty acids and weight of animals were similar in the four groups. The weight of the heart was significantly higher in the GK group compared to the three other groups (*p* < 0.001). However, the heart weight to body weight ratio was increased in GK (*p* < 0.001), GK-P (*p* < 0.01), and GK-RSV (*p* < 0.01) versus CTRL. The heart weight to body weight ratio was decreased in GK-RSV (*p* < 0.05) and GK-P (*p* < 0.01) in comparison to GK, indicating decreased cardiac hypertrophy by RSV treatment.

### 3.2. Effect of Resveratrol on Tolerance to Ischemia-Reperfusion Injury

#### 3.2.1. Myocardial Function

Myocardial function (Figure 1A) was impaired in GK, GK-P, and GK-RSV compared with CTRL in baseline conditions (*p* < 0.001 GK and GK-P vs. CTRL; *p* < 0.01 GK-RSV vs. CTRL). RSV did not improve cardiac function in GK-RSV in comparison to CTRL in baseline conditions. After ischemia, GK and GK-P groups presented a higher sensitivity to ischemia-reperfusion injury since myocardial function was significantly impaired compared with CTRL (*p* < 0.001) and the percentage of recovery (Figure 1B) was significantly decreased (respectively *p* < 0.001 and *p* < 0.01 vs. CTRL). Interestingly, GK-RSV rats had a better tolerance to ischemia-reperfusion injury than GK and GK-P rats, with an improvement of cardiac function up to CTRL values.

#### 3.2.2. Myocardial Energy Metabolism

##### ^31^P MRS

Kinetics of PCr, ATP, Pi, and pHi during the experimental time course are shown in Figure 2. No difference was found between groups in baseline conditions and during ischemia for PCr (Figure 2A) and ATP (Figure 2B) heart contents. However, during reperfusion, PCr and ATP heart contents were significantly decreased in GK and GK-P when compared with CTRL (*p* < 0.05). RSV restored PCr and ATP contents to control values during reperfusion. In baseline conditions, Pi (Figure 2C) was not different between groups. During ischemia, Pi was significantly higher in GK and GK-P in comparison with CTRL (respectively *p* < 0.001 and *p* < 0.05). RSV was able to prevent the increase in Pi in GK-RSV rats. No statistical difference was found between CTRL and GK-RSV, and Pi was significantly lower in GK-RSV vs. GK (*p* < 0.001). Finally, pHi (Figure 2D) was identical between groups in baseline conditions. During ischemia, pHi was significantly decreased in GK (*p* < 0.01 vs. CTRL). During reperfusion, pHi was significantly decreased in GK and GK-P compared with GK-RSV (respectively *p* < 0.01 and *p* < 0.05). RSV treatment restored pHi in GK-RSV to control values.

##### Biochemical Analysis in Freeze-Clamped Hearts

Considering the improvements made by the RSV on high-energy compound contents during ex vivo experiments, we carried out additional biochemical analyses in freeze-clamped hearts. First, a total pool of PCr, creatine, ATP, and total adenine nucleotides (TAN) were assessed using HLPC as shown in Figure 3A. PCr was significantly decreased in GK and GK-P in comparison to CTRL (*p* < 0.05). RSV restored PCr heart content in GK-RSV, which was significantly different compared with GK (*p* < 0.01) and GK-P (*p* < 0.05). Creatine was not different between CTRL, GK, and GK-P groups. RSV treatment increased creatine heart content in GK-RSV versus GK (*p* < 0.001) and GK-P (*p* < 0.05). The sum of creatine and phosphocreatine was significantly increased in GK-RSV in comparison to GK (*p* < 0.001) and GK-P (*p* < 0.01). ATP was significantly decreased in GK and GK-P in comparison to CTRL (*p* < 0.01). RSV increased ATP content in GK-RSV versus GK and GK-P (respectively *p* < 0.01 and *p* < 0.05). TAN was significantly decreased in GK and GK-P in comparison to CTRL (*p* < 0.01). RSV restored TAN in GK-RSV, which was increased in comparison to GK (*p* < 0.01) and GK-P (*p* < 0.05). These results are in line with energy metabolism measured by ^31^P MRS. AMP, ADP, and energy charge results are shown in the Appendix A. No statistical difference was found between groups for AMP content. ADP content was significantly decreased only in GK versus CTRL (*p* < 0.05). RSV increased ADP content but it did not reach statistical difference. No difference was found between groups for energy charge. Second, citrate synthase activity (Figure 3B) was also assessed in freeze-clamped hearts. Citrate synthase activity was significantly increased by RSV treatment in GK-RSV in comparison to the other groups (*p* < 0.0001). Together these results indicate that RSV could improve cardiac mitochondrial function in type 2 diabetic female rats.

#### 3.2.3. Coronary Flow and Expression of Proteins Involved in NO Pathway

Before ischemia (Figure 4A), CF was significantly decreased in GK-P in comparison to GK-RSV (*p* < 0.05). During reperfusion (Figure 4B), CF was significantly impaired in GK and GK-P in comparison to CTRL (*p* < 0.01). Treatment with RSV maintained CF during reperfusion to control values in GK-RSV.

Complementary to the coronary flow measurement, we assessed the expression of eNOS, SIRT1, Akt and P-Akt proteins involved in the NO pathway in freeze-clamped hearts. Expression of eNOS, Akt, PAkt (Ser 473), and SIRT1 protein is shown in Figure 5. eNOS protein expression was significantly increased in GK-RSV in comparison to the three other groups (*p* < 0.05 vs. CTRL and GK-P; *p* < 0.01 vs. GK). Akt protein was similarly expressed in the four groups. The phosphorylated form of Akt was significantly increased in GK-RSV vs. CTRL and GK-P (*p* < 0.05). SIRT1 was increased in GK-RSV compared to the other groups (*p* < 0.05). These results suggest an improvement of NO pathway by RSV leading to higher coronary flow during reperfusion. iNOS was not expressed in the four groups (data not shown).

##### Oxidative Stress

SIRT3 protein expression and MDA heart content were not different between groups, with no effect of RSV (Appendix A).

## 4. Discussion

The main objective of this study was to investigate the potential protective effects of RSV on high-energy compounds during ischemia-reperfusion injury in type 2 diabetic female rat hearts. We found in GK rat hearts a lower tolerance to ischemia-reperfusion injury, characterized by impaired energy metabolism and associated with a decrease in functional recovery and coronary flow. Eight-week treatment with a low dose of RSV was able to protect the heart from the loss of energetic compounds during reperfusion and to improve cardiac function and coronary flow. Biochemical analyses confirmed the positive effects of RSV on ATP and PCr, as well as TAN, creatine, and citrate synthase activity, which are indicators of mitochondrial function. Moreover, the improvement of coronary flow during reperfusion by RSV was associated to increased eNOS, SIRT1, and P-Akt protein expression in GK rat hearts.

RSV has been associated to the French paradox, which reflects the lower incidence and mortality by CV disease in the French population, as a link with daily consumption of red wine [32]. As specified in the introduction, a dose of less than 1 mg/kg/day may not have cardiovascular effects [18], while a high dose may damage the heart during an ischemia-reperfusion injury [24]. Thus, the dose used in our study seems to be a good compromise.

The Goto-Kakizaki rat is one of the best characterized animal models of spontaneous type 2 diabetes [33] presenting cardiac insulin resistance and CV complications [34]. Here we found cardiac hypertrophy and basal cardiac dysfunction in GK vs. CTRL due to the decrease of both developed pressure and heart rate. The modification of high energy compounds does not explain the impaired cardiac function found in baseline conditions. Interestingly, the alteration of excitation–contraction coupling [35] and the downregulation and upregulation of multiple genes, such as Trpc6 or Ryr2, involved in the activity of the sinoatrial node [36] have been previously reported in GK rats and could explain the impairment in myocardial function shown here. RSV had no effect on cardiac function prior to ischemic insult, as previously reported by Robich et al. [37]. However, 1 mg/kg/day of RSV decreased cardiac hypertrophy and improved the myocardial tolerance to ischemia-reperfusion injury. Recently, Bagul et al. showed cardiac hypertrophy with increased cardiac cell size in rats under a high-fat diet, with a reverse effect of RSV administered in the food at 10 mg/kg/day for 8 weeks [19]. Lin et al. pointed out the decrease of atrial natriuretic peptide and TGF1β related to reduced infarct size in animals treated with RSV by intraperitoneal injection for 4 weeks [18]. RSV has also been shown to reduce pro-hypertrophic markers such as ANP, BNP, and β-MHC, and improve redox balance by increasing SOD [13] in streptozotocin (STZ) and high-fat model of type 2 diabetes. Interestingly, placebo treatment also showed a decrease in cardiac hypertrophy in type 2 diabetic rats. Placebo treatment (ethanol 1‰) may have an effect on cardiac hypertrophy, as suggested by Ninh et al. in a rodent model of pressure overload with cardiac hypertrophy [38]. Moreover, Miyamae et al. also showed a higher myocardial tolerance to ischemia-reperfusion injury in animals treated with ethanol [39]. Nonetheless, the authors used up to 20% of ethanol in the drinking water, which might explain why we did not see an effect on the tolerance to ischemia-reperfusion injury in our study.

Myocardial tolerance to ischemia-reperfusion injury was impaired in type 2 diabetic GK rats and was associated with altered energy metabolism, characterized by a decrease in high energy compound contents. Indeed, mitochondrial dysfunction has been widely suggested to explain the mechanisms involved in heart failure of diabetic patients [40]. Studies on type 2 diabetic animals also showed decreased expression of mitochondrial respiratory chain complexes, and mitochondrial biogenesis through PGC1α [41]. Interestingly, a previous study on the GK model showed impaired cardiac function during ischemia-reperfusion injury, without alteration of energy metabolism in male gender [27]. In addition, Billimoria et al. showed a decrease of mitochondrial respiration in diabetic STZ rat hearts with a higher impairment in female than in male [42], unlike female GK here. Remarkably, RSV improved high energy compounds during reperfusion in GK-RSV rats and this observation may explain the better myocardial functional recovery. ATP and PCr were significantly increased in GK-RSV rats, up to control values. In parallel RSV prevented the high increase in Pi during ischemia and decrease in pHi during reperfusion. Consistent with these results, HPLC analysis in the cardiac tissue highlighted the restoration in ATP and PCr heart contents in GK-RSV at the end of reperfusion. In addition, we showed a preservation in the pool of creatine and TAN, crucial for ATP and PCr synthesis, in type 2 diabetic animals under RSV after ischemic insult. Here, RSV treatment also increased CS activity in GK rat hearts, as recently reported by Lagouge et al. in mice treated orally with a dose of 400 mg/kg/day RSV, indicating enhanced mitochondrial enzymatic activity [43]. Consequently, taken together, our results suggest that the RSV-induced cardioprotection against ischemia-reperfusion injury in type 2 diabetes could be associated to better mitochondrial functioning. In the literature, mitochondrial function has been shown to be improved by RSV via increase in mitochondrial DNA, biogenesis mitochondrial factor PGC1α [13], Nrf-1, and Tfam mRNA expression [44], and decrease in the opening of mitochondrial transition pore [21]. Further studies need to be performed to elucidate the molecular mechanisms involved in the RSV-induced mitochondrial protection.

On the other hand, the expression of the SIRT3 protein, a mitochondrial sirtuin involved in mitochondrial function and oxidative stress [45], was the same in all groups. This result is consistent with MDA heart content which was also similar in all groups. Thus, the improvement of energy metabolism by RSV was independent from SIRT3 and oxidative stress.

Multiple studies have shown the effect of RSV, a well-known SIRT1 activator, on endothelial function [14,15,16,17]. Here, we assessed coronary flow and expression of proteins involved in NO pathway as indicators of endothelial function. During reperfusion, coronary flow was altered in both GK and GK-P versus CTRL. No difference was shown in eNOS, Akt, P-Akt, and SIRT1 expression, between CTRL and GK rats, although previous studies reported decreased eNOS expression in type 2 diabetes. The literature is still inconclusive concerning the expression of eNOS in type 2 diabetes. Indeed, some studies reported a decrease [46] while others showed an increase [27] in eNOS expression. Interestingly, RSV was able to fully restore the coronary flow during reperfusion and to significantly increase the expression of eNOS, P-Akt, and SIRT1 proteins in GK rat hearts. Previously, Huang et al. showed that RSV increased the expression of P-Akt and eNOS in the thoracic aorta of rats under high-fat diet [47]. In type 2 diabetic db/db mice, RSV also enhance cardiac NO production and eNOS protein expression [48]. More generally, RSV has also been shown to enhance NO production, increase NOS expression and activity, prevent eNOS uncoupling and increase NO bioavailability [9]. In fact, increasing NO production and bioavailability via eNOS is one of the mechanisms involved in cardioprotection against ischemia-reperfusion [49]. Then, exploring the phosphorylated form of eNOS, eNOS uncoupling or NO availability could help us understand the higher coronary flow reported in GK-RSV rats during reperfusion. Remarkably, RSV increased tolerance to ischemia-reperfusion injury independently from glycemic improvements. Indeed, we did not observe any effect on glycemia with a low-dose of 1 mg/kg/day. Some studies present RSV as a new potential anti-diabetic treatment when used at high dose [13,50]. The mechanisms involved might go through the increase in GLP-1 secretion [50], beta cell insulin secretion, beta cell gene expression, or improvement of insulin sensitivity [51]. At this point, it is important to remind that the GK model presents mild hyperglycemia, which could explain why effects of RSV might go unnoticed. Therefore, we may suppose an estrogen-like effect of RSV on mitochondrial and endothelial pathways, which could improve tolerance to ischemia-reperfusion injury, independently from glycemic control. Estrogens have positive effects on vessels by improving vasorelaxation [52] and on key regulators of energy metabolism and mitochondrial biogenesis (PGC1α) [13]. Moreover, RSV has an estrogen-like effect by activating estrogen receptors at nuclear and extracellular levels [53]. Recently, RSV has shown better effects on metabolic parameters in female controls than in ovariectomized female rats [54]. It would, therefore, be interesting to assess the effects of RSV on ovariectomized female GK rats to better understand the involvement of hormones in cardiovascular RSV effects.

In conclusion, RSV had a protective effect against ischemia-reperfusion injury via increased high energy compound contents and eNOS-SIRT1 expression in type 2 diabetic female rat heart. We believe our results could contribute to a better understanding of the mechanisms involved in RSV-induced cardioprotection. As type 2 diabetic women present a high risk of mortality by myocardial infarction, low dose of RSV supplementation could be an interesting way to improve myocardial infarction survival. Indeed, mitochondrial and endothelial dysfunctions have been reported in the type 2 diabetic patients, with a decrease in PCr/ATP ratio in the heart and a high rate of coronary artery diseases. Thus, RSV presents high potential for preventing and treating cardiovascular complications of type 2 diabetic women.

## Figures and Tables

**Figure 1 nutrients-11-00105-f001:**
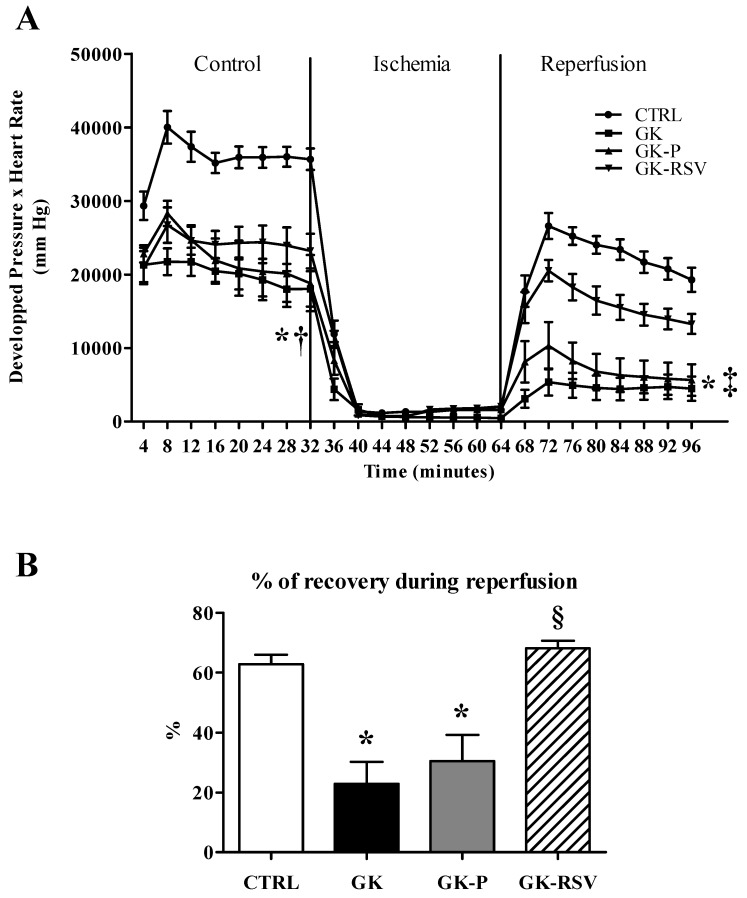
Myocardial function evaluated by the product of developed pressure and heart rate during the experimental time course (**A**) and % of recovery during reperfusion (**B**). Results are expressed as means ± SEM. Two-way ANOVA was performed to observe the effect of group and time. * *p* < 0.001 GK and GK-P vs. CTRL, † *p* < 0.01 GK-RSV vs. CTRL, ‡ *p* < 0.01 GK and GK-P vs. GK-RSV, and § *p* < 0.001 vs. GK and GK-P.

**Figure 2 nutrients-11-00105-f002:**
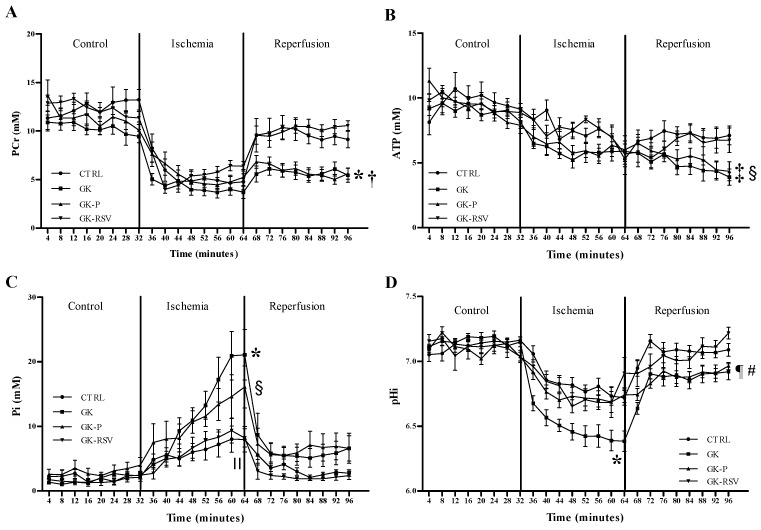
Kinetics of phosphocreatine (PCr) (**A**), ATP (**B**), Pi (**C**), and intracellular pH (pHi) (**D**) during the experimental time course in rat hearts. Data are expressed as means ± SEM. Two-way ANOVA was performed to observe the effect of group and time. * *p* < 0.01 GK vs. CTRL, † *p* < 0.01 GK-P vs. CTRL, ‡ *p* < 0.05 GK vs. CTRL, § *p* < 0.05 GK-P vs. CTRL, II *p* < 0.001 GK-RSV vs. GK, ¶ *p* < 0.05 GK-P vs. GK-RSV, and # *p* < 0.01 GK vs. GK-RSV.

**Figure 3 nutrients-11-00105-f003:**
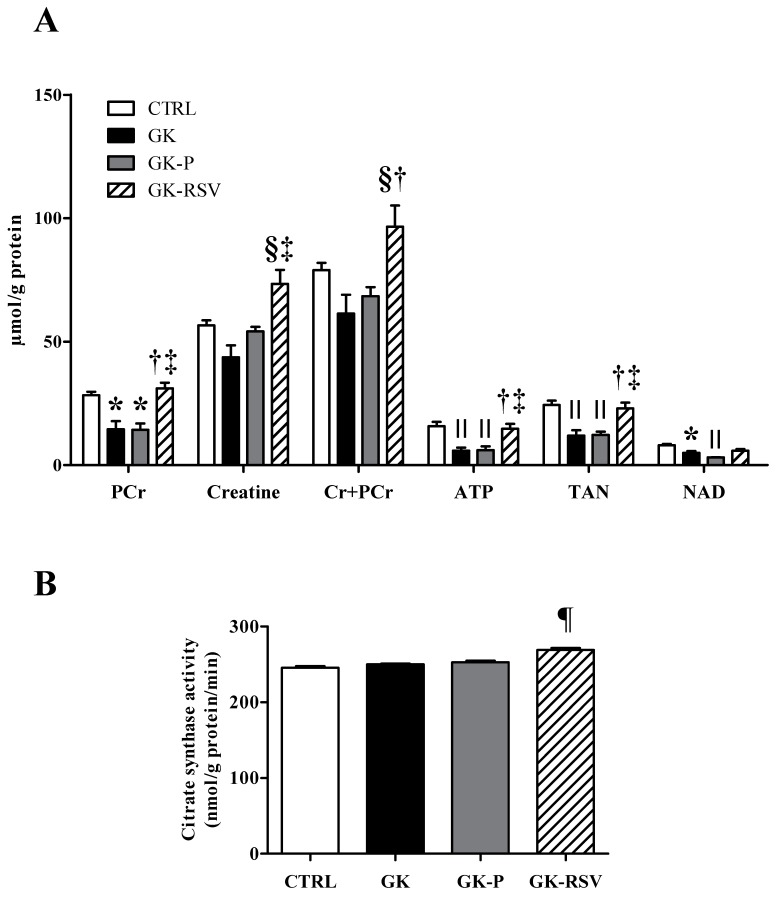
Total pool of phosphocreatine, creatine, PCr + Cr, ATP, total adenine nucleotides (TAN) (**A**) and citrate synthase activity (**B**) in rat hearts. Data are expressed as means ± SEM and one-way ANOVA was used to compare the groups. * *p* < 0.05 vs. CTRL, † *p* < 0.01 vs. GK, ‡ *p* < 0.05 vs. GK-P, § *p* < 0.001 vs. GK; II *p* < 0.01 vs. CTRL; and ¶ *p* < 0.0001 vs. CTRL, GK, and GK-P.

**Figure 4 nutrients-11-00105-f004:**
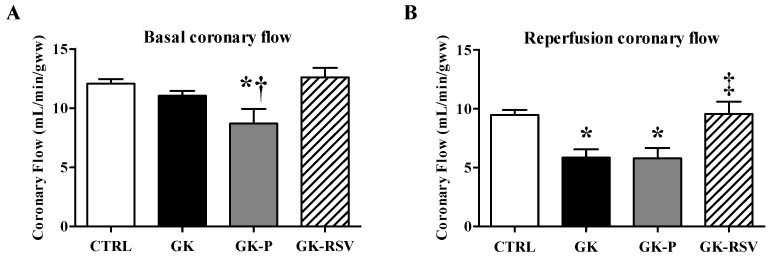
Baseline coronary flow evaluated at 20 min (**A**) and reperfusion coronary flow evaluated at 80 min (**B**). Data are expressed as means ± SEM and one-way ANOVA was used to compare the groups. * *p* < 0.05 vs. CTRL, † *p* < 0.01 vs. GK-RSV, and ‡ *p* < 0.05 vs. GK and GK-P.

**Figure 5 nutrients-11-00105-f005:**
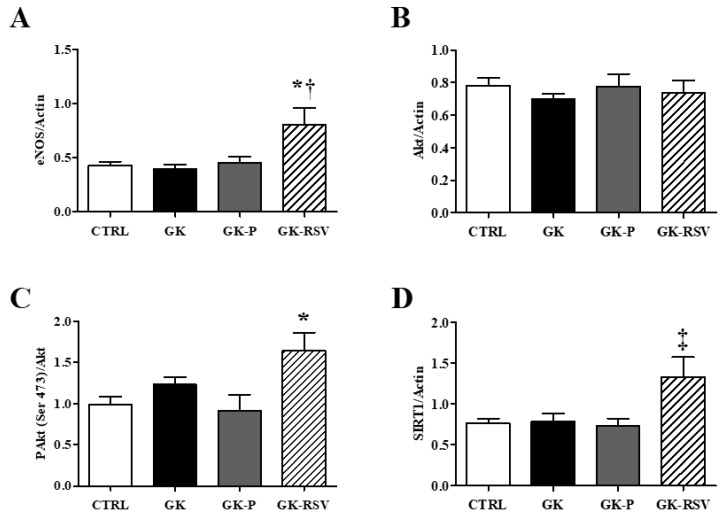
Protein expression of eNOS (**A**), Akt (**B**), PAkt (Ser 473) (**C**), and SIRT1 (**D**). Representative western blot of eNOS (**E**), Akt and its phosphorylated form (**F**), and SIRT1 (**G**). Data are expressed as means ± SEM and one-way ANOVA to compare the groups. * *p* < 0.05 vs. CTRL and GK-P; † *p* < 0.01 vs. GK; and ‡ *p* < 0.05 vs. CTRL, GK, and GK-P.

**Table 1 nutrients-11-00105-t001:** Physiological parameters of experimental animals.

	CTRL	GK	GK-P	GK-RSV
Glycemia (g/L)	1.64 ± 0.07	2.46 ± 0.09 *	2.52 ± 0.15 *	2.49 ± 0.07 *
Free Fatty Acids (mM)	0.21 ± 0.05	0.18 ± 0.02	0.17 ± 0.04	0.16 ± 0.04
Body Weight (g)	289.4 ± 6.6	284.6 ± 4.4	267.5 ± 5.7	269.7 ± 6.4
Heart Weight (g)	0.85 ± 0.02	1.05 ± 0.03 * ^† ‡^	0.89 ± 0.02	0.91 ± 0.02
(Ratio Heart/Body Weight) × 1000	2.95 ± 0.09	3.69 ± 0.06 * ^II ¶^	3.32 ± 0.06 ^§^	3.38 ± 0.09 ^§^

Data are expressed as means ± SEM. One-way ANOVA test was used for all the parameters. * *p* < 0.0001 vs. CTRL; † *p* < 0.0001 vs. GK-P; ‡ *p* < 0.0001 vs. GK-RSV; § *p* < 0.01 vs. CTRL; II *p* < 0.01 vs. GK-P; ¶ *p* < 0.05 vs. GK-RSV.

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
