# Peer review of "Protective Effect of Resveratrol against Ischemia-Reperfusion Injury via Enhanced High Energy Compounds and eNOS-SIRT1 Expression in Type 2 Diabetic Female Rat Heart"

_nutrients, 2019, doi:10.3390/nu11010105_

Round 1
Reviewer 1 Report
Comments to the Author
The manuscript examines RSV induced cardioprotection against ischemia-reperfusion injury in type 2 diabetic female rats. However, although the findings are valuable, they are not well described or discussed and even overstated or misinterpreted at times. The manuscript needs major revisions to remove the multiple inaccuracies and inconsistencies, improve the logical flow, and reformat it into a clear background à hypothesis à results à conclusions form. Specific major and minor comments are included.
Major comments
Abstract
The abstract creates an impression that the study describes random outcomes. Clear hypothesis and logical explanation of the results are often lacking throughout the paper, although they are essential for the scientific method.
Introduction
line 57-58 ……low oral dose justification not given anywhere in the manuscript. Provide proper justification line 59-60…. Mentioned multiparameter approach myocardial and endothelial function. What endothelial function measured? I could not find anything.
Line 61-63… don’t use the phrase “the first study” and effect of RSV on mitochondria…. What parameter done in this manuscript related to mitochondria.
Materials and methods
Line 149: Technically manuscript deals with few proteins say for example eNOS, SIRT1, Akt and its phosphorylated form, but the title written pathway; avoid overstating.
Line 140: A piece of heart tissue ….. How much weight? And from which part of the heart or chamber? Needs to clarify.
Line154 and 164: Why two different concentrations of protein used and from my experience 90ug protein is too much saturation concentration for internal control actin in the heart?
Results, Figures, Table and discussion
Table 1. What was the explanation there is no change in glycemia in the RSV group? Then how could one expect the diabetic cardiometabolic complications attenuated by RSV? And explain the discussion accordingly.
Line 187-188: Cardiac hypertrophy attenuated is it true? On what basis you are claiming this statement, with heart weight even though it is not altered significantly in the RSV group.
Figure 5. I was wondering why there is no representative blots and actin is misspelled actine in the axis. pAkt must be normalized with Akt, not with actin and mention it is Ser473 in figures and material methods section also.
The discussion as a whole lacks logical clarity and needs to be rewritten to clearly discuss how the described findings fit in the hypothesis and what their implications for future are.
Minor comments
Materials methods, subtitles next to the first letter there was a dot is it typeset issue or it was in original manuscript itself.
Animals, chemical, antibody company name, and location must be mentioned throughout the manuscript.
Many phrases were not grammatically accurate and need to be rephrased. The manuscript needs profession language proofreading.
Author Response
Nutrients
Manuscript ID: nutrients-405198 Revision 1
Reviewer 1
We thank this Reviewer for his helpful comments and believe that our manuscript has been improved markedly following his suggestions to which we respond more fully as follows:
Major comments
1) The abstract creates an impression that the study describes random outcomes. Clear hypothesis and logical explanation of the results are often lacking throughout the paper, although they are essential for the scientific method.
We agree and we modified the abstract and the paper to provide a clear hypothesis and logical explanation of the results as suggested.
The Materials and Methods, Results and Discussion sections have been now reorganized, in order to better present the main objectives and hypothesis of our work, and to highlight more clearly the effects of resveratrol.
2) line 57-58 ……low oral dose justification not given anywhere in the manuscript. Provide proper justification
The literature reports doses ranging from 0.1 mg/kg/day (lin et al., doi 10.1016/j.lfs.2008.06.016) to 500 mg/kg/day (Meng et al., Am J Transl Res. 2016; 8(6): 2641–2649) and even 4g/kg/day (Shah et al., 10.1113/JP271133). Here we choose a “low-dose of RSV at 1 mg/kg/day based on Rocha et al. and Lin et al. studies. Indeed, Rocha et al. (doi 10.1016/j.fct.2009.03.010) chose a dose of 1 mg/kg based on the actual wine consumption in occidental countries, the kinetics and bioavailability of resveratrol in the body. Lin et al. (doi 10.1016/j.lfs.2008.06.016) study showed that a lower dose (0.1 mg/kg) was not sufficient to induce beneficial effects on the heart, while a dose of 1 mg/kg/day improved cardiac function. On the other hand, higher doses (25 mg/kg/day) showed negative effects on infarct size (Dudley et al., doi 10.1021/jf3008597).
Justification of low-dose RSV was added in the introduction of the manuscript (page 2, lines 57-65).
3) line 59-60…. Mentioned multiparameter approach myocardial and endothelial function. What endothelial function measured? I could not find anything.
This is indeed a very good point. In our study we used the coronary flow and the expression of proteins involved in the NO pathway as indicators of endothelial function. The term ‘endothelial function” might be too confusing and not enough specific, as we did not explore the endothelium-dependent and –independent vasodilatations. We replaced “endothelial function” by the specific terms of coronary flow and expression of proteins involved in the NO pathway throughout the paper.
4) Line 61-63… don’t use the phrase “the first study” and effect of RSV on mitochondria…. What parameter done in this manuscript related to mitochondria.
The phrase was removed.
We evaluated high energy compounds (ATP, PCr, Pi, TAN) and creatine using 31P MRS and HPLC methods, as well as citrate synthase activity, which are indicators of mitochondrial function. As for endothelial function, we changed the term “mitochondrial function” for more specific term, in the title and throughout the paper.
5) Line 149: Technically manuscript deals with few proteins say for example eNOS, SIRT1, Akt and its phosphorylated form, but the title written pathway; avoid overstating.
The title was changed as suggested (page 4, line 176; page 8, line 285).
6) Line 140: A piece of heart tissue …. How much weight? And from which part of the heart or chamber? Needs to clarify.
60 mg of the left ventricle were homogenized in RIPA lysis buffer and centrifuged at 14,000 rpm for 15 min at 4 °C. Total protein concentration in supernatant was determined using the Pierce BCA protein assay kit. Equal amounts of protein were then separated on polyacrylamide gel electrophoresis. This information was added in the Materials and Methods section (page 4, line 178).
7) Line154 and 164: Why two different concentrations of protein used and from my experience 90ug protein is too much saturation concentration for internal control actin in the heart?
Higher protein concentrations have been previously used in the literature for determination of eNOS expression. Gödecke et al. (Circ Res. 1998 Feb 9;82(2):186-94) used 200 µg of protein and Caus et al. (J Heart Lung Transplant. 2003 Feb;22(2):184-91) used 130 µg of protein. In these studies, eNOS was expressed in arbitrary units. The protocol used here to determine the expression of eNOS is a modified protocol inspired from Gödecke et al. with 90 µg of protein per well, to express results according to Actin. Protein concentrations for both eNOS and PAkt have been validated in a previous study on GK rat hearts (Nutr Metab (Lond), 2017 Jan 13;14:6. doi: 10.1186/s12986-016-0157-z).
8) Table 1. What was the explanation there is no change in glycemia in the RSV group? Then how could one expect the diabetic cardiometabolic complications attenuated by RSV? And explain the discussion accordingly.
This is an excellent question. Indeed, it is surprising to observe no effect of RSV on glycemia while RSV improved cardiometabolic complications. It is important to remind that the GK model presents mild hyperglycemia which could explain why the effects of RSV on glycemia might go unnoticed. We may suppose an estrogen-like effect of RSV on mitochondrial and endothelial pathways which could improve tolerance to ischemia-reperfusion injury, independently from glycemic control. This point has been now included in the Discussion (page 12, lines 404-418).
9) Line 187-188: Cardiac hypertrophy attenuated is it true? On what basis you are claiming this statement, with heart weight even though it is not altered significantly in the RSV group.
GK rats present a significant increase in heart weight in comparison to the three other groups. However, we used the heart weight to body weight ratio as an index of cardiac hypertrophy, rather than the heart weight itself. In comparison to CTRL group, GK rats present a higher ratio indicating cardiac hypertrophy. RSV treatment decreased significantly heart to body weight ratio in GK-RSV in comparison to GK, that is why we state that RSV treatment attenuated cardiac hypertrophy. This point is now more clearly explained in the Results (page 5, lines 208-213).
10) Figure 5. I was wondering why there is no representative blots and actin is misspelled actine in the axis. pAkt must be normalized with Akt, not with actin and mention it is Ser473 in figures and material methods section also.
Representative blots have been added to figures under histograms to illustrate our results (Figure 5F, 5G, 5G). The term “Actin” has been corrected. We normalized P-Akt with Akt as requested. Finally, we mention Ser473 for phosphorylated form of Akt (page 10).
11) The discussion as a whole lacks logical clarity and needs to be rewritten to clearly discuss how the described findings fit in the hypothesis and what their implications for future are.
This comment was very helpful. We believe that we clarified the Discussion to fit the hypothesis of our study. In perspective, we also connect our results to potential clinical applications (page 13, lines 420-428).
Minor comments
1) Materials methods, subtitles next to the first letter there was a dot is it typeset issue or it was in original manuscript itself.
It was not in the original manuscript, so it was probably an issue that appeared when the text was formatted. We have corrected it.
2) Animals, chemical, antibody company name, and location must be mentioned throughout the manuscript.
We added company names and locations when it was forgotten in the original manuscript.
3) Many phrases were not grammatically accurate and need to be rephrased. The manuscript needs profession language proofreading.
We have read the whole manuscript and revised English grammar and word usage.

Reviewer 2 Report
Please see my comments below:
1) Delete the fullstop in the title.
2) Line 54: Please provide a little bit more detail on this.
3) Line 171: Please provide more details for the methods.
4) Line 185: use "was" instead of "is".
5) Results: the authors should present the main results first based on their objectives followed by secondary objectives.
6) Discussion: how does this related to humans based on the rat models? This is not clearly explained.
7) References: please format the style according to the Nutrients format.
8) Please add a subsection about materials and equipment under the Methods.
9) line 65: Please correct to "Animals".
10) line 196: Please explain "no effect".
Author Response
Nutrients
Manuscript ID: nutrients-405198 Revision 1
Reviewer 2
Many thanks to this reviewer for his helpful comment, which we have addressed in the manuscript. Point by point answers are provided below.
1) Delete the fullstop in the title.
The full stop has been deleted in the title.
2) Line 54: Please provide a little bit more detail on this.
In the literature, doses of RSV can be very different depending on the study, ranging from 0.1 mg/kg/day (lin et al., doi 10.1016/j.lfs.2008.06.016) to 500 mg/kg/day (Meng et al., Am J Transl Res. 2016; 8(6): 2641–2649) and even 4g/kg/day (Shah et al., doi 10.1113/JP271133). Here we choose a low-dose of RSV at 1 mg/kg/day based on Rocha et al. and Lin et al. studies. Indeed, Rocha et al. (doi 10.1016/j.fct.2009.03.010) chose a dose of 1 mg/kg based on the actual wine consumption in occidental countries, the kinetics and bioavailability of resveratrol in the body. Lin et al. (doi 10.1016/j.lfs.2008.06.016) study showed that a lower dose (0.1 mg/kg) was not sufficient to induce beneficial effects on the heart, while a dose of 1 mg/kg/day improved cardiac function. On the other hand, higher doses (25 mg/kg/day) showed negative effects on infarct size (Dudley et al., doi 10.1021/jf3008597).
These explanations were now added in the manuscript (page 2, lines 57-65).
3) Line 171: Please provide more details for the methods.
Lipid peroxidation was determined by the reaction of MDA with thiobarbituric acid (TBA) to form a colorimetric (532 nm)/fluorometric (λex = 532/λem = 553 nm) product, proportional to the concentration of MDA.
Details were added in the manuscript (page 5, lines 194-196).
4) Line 185: use "was" instead of "is".
Corrected.
5) Results: the authors should present the main results first based on their objectives followed by secondary objectives.
The Materials and Methods, Results and Discussion sections have been now reorganized, in order to better present the main objectives and hypothesis of our work, and to highlight more clearly the effects of resveratrol.
6) Discussion: how does this related to humans based on the rat models? This is not clearly explained.
In perspective, we connect our results to potential clinical applications (page 13, lines 420-428).
7) References: please format the style according to the Nutrients format.
Correction was performed.
8) Please add a subsection about materials and equipment under the Methods.
Following your recommendation, a subsection called “Materials and antibodies” was added at the beginning of Materials and Methods section (page 2, lines 75-92).
9) line 65: Please correct to "Animals".
It was not in the original manuscript, so it was probably an issue that appeared when the text was formatted. We have corrected it.
10) line 196: Please explain "no effect".
Myocardial function (Figure 1A) was impaired in GK, and GK-P compared with CTRL in baseline conditions (p<0.001 GK and GK-P vs CTRL; p<0.01 GK-RSV vs. CTRL). RSV did not improve cardiac function in GK-RSV in comparison to CTRL in baseline conditions.
This explanation was added in the text (page 5, lines 220-222).

Round 2
Reviewer 1 Report
I appreciate the authors for the considerable changes and current form is scientifically sound and interest to the readers.